# RIGID: A Training-Free and Generator-Agnostic Framework for Robust AI-Generated Image Detection

**Zhiyuan He**                                                     *zyhe@cse.cuhk.edu.hk*
*Department of Computer Science and Engineering*
*The Chinese University of Hong Kong*

**Pin-Yu Chen**                                                   *pin-yu.chen@ibm.com*
*IBM Research*

**Tsung-Yi Ho**                                                   *tyho@cse.cuhk.edu.hk*
*Department of Computer Science and Engineering*
*The Chinese University of Hong Kong*

**Reviewed on OpenReview:** *https://openreview.net/forum?id=NBkBI2Zjlm*

## Abstract

The rapid advances in generative AI models have empowered the creation of highly realistic images with arbitrary content, raising concerns about potential misuse and harm, such as deepfakes. Current research focuses on training detectors using large datasets of generated images. However, these training-based solutions are often computationally expensive and show limited generalization to unseen generated images. In this paper, we propose a *training-free* method to distinguish between real and AI-generated images. We first observe that real images are more robust to tiny noise perturbations than AI-generated images in the representation space of vision foundation models. Based on this observation, we propose RIGID, a training-free and generator-agnostic method for robust AI-generated image detection. RIGID is a simple yet effective approach that identifies whether an image is AI-generated by comparing the representation similarity between the original and the noise-perturbed counterpart. Our comprehensive evaluation demonstrates RIGID's practical effectiveness. On the IMAGENET and LSUN-BEDROOM averages, RIGID improves AP over AEROBLADE by 26.07 and 28.49 points, respectively. Remarkably, RIGID performs comparably to training-based methods, particularly on out-of-domain data. Additionally, RIGID maintains competitive performance across a broad range of generation techniques and demonstrates strong resilience to common image corruptions. *Code and data are available at https://github.com/IBM/RIGID.*

## 1 Introduction

In recent years, deep learning has revolutionized image generation, enabling the creation of highly realistic images. Platforms such as Stable Diffusion (Rombach et al., 2022b) and Midjourney (Midjourney, 2022) allow users to generate arbitrary content through text prompts. However, these advanced Generative AI (GenAI) applications are accompanied by amplified risks and concerns about misuse, such as deepfakes. Some prompt-based jailbreak techniques (Chin et al., 2024; Tsai et al., 2023; Yang et al., 2024) can bypass platforms' safeguards and generate inappropriate content, highlighting the urgent need for practical and reliable AI-generated image detection.

For AI-generated image detection, a common practice is to design a detector that learns to distinguish between real and generated images. Early research (Frank et al., 2020; Dzanic et al., 2020; Chandrasegaran et al., 2021) discovered that the upsampling process in Generative Adversarial Network (GAN (Goodfellow

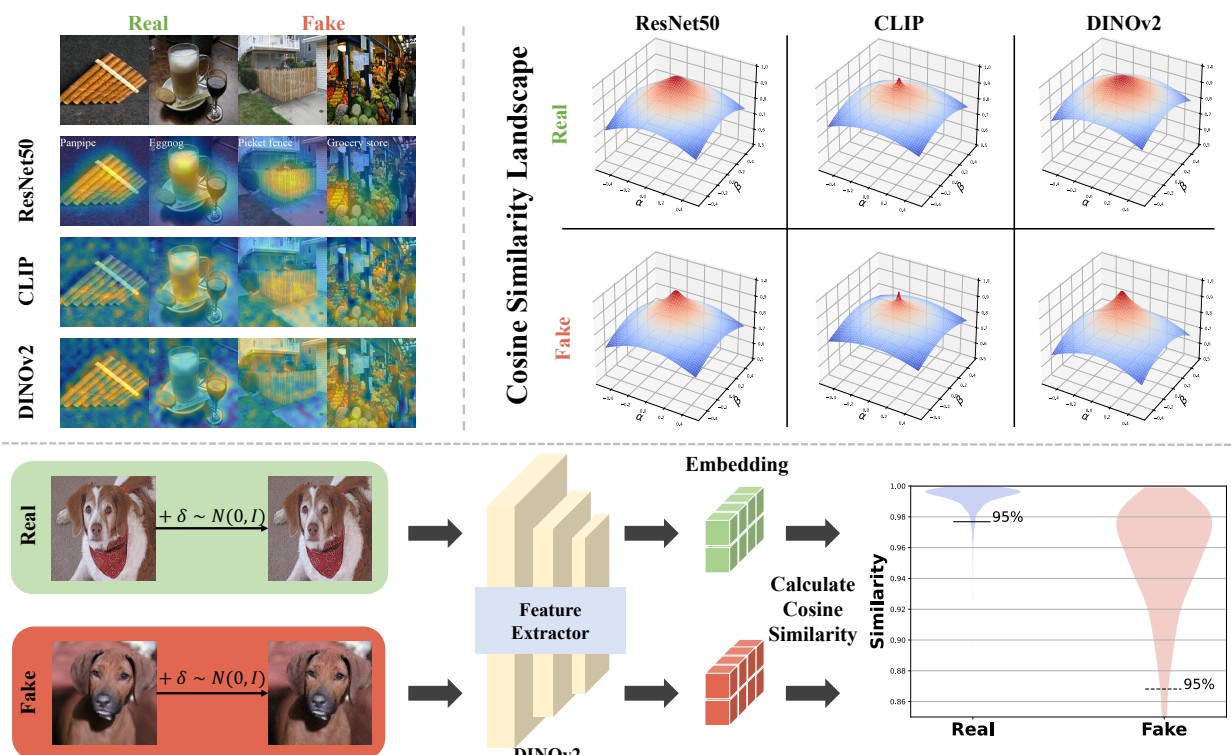

Figure 1: Overview of **RIGID. Upper left:** visualization of the attention range of different models for real images and AI-generated (fake) images by GradCAM (Selvaraju et al., 2017). CLIP and DINOV2 attend better to global context than ResNet 50. **Upper right:** visualization of the cosine similarity landscape for real and AI-generated images by plotting the interpolation of two random directions in the image pixel space with coefficients $\alpha$ and $\beta$. See details of the landscape visualization in the Appendix. We find that on DINOv2, real and AI-generated images exhibit distinct sensitivity results. **Bottom:** the framework of RIGID. RIGID uses a pretrained feature extractor to compute the pairwise cosine similarity on the original and noise-perturbed images for AI-generated image detection. The entire detection process is training-free, generator-agnostic, and efficient. See Sec. 3.1 for details.

et al., 2020)) leaves periodic artifacts in the spatial or frequency domain of the generated images, allowing for effective detection of low-quality generated images by checking these specific traces. However, synthetic artifacts have been weakened with advances in generation methods (Corvi et al., 2023). This has led to the development of numerous training-based detection methods, which learn common features of generated images by training on large datasets of real and fake images. Wang et al. (Wang et al., 2020) show that a deep neural network (DNN) classifier trained on images from a single GAN can surprisingly generalize to images from unseen GANs. Gragnaniello et al. (Gragnaniello et al., 2021) enhance detection performance by using extensive data augmentations. Corvi et al. (Corvi et al., 2023) train a classifier on images generated by Latent Diffusion Model (LDM (Rombach et al., 2022a)). Ojha et al. (Ojha et al., 2023) train a simple linear classifier on features extracted from the pretrained CLIP (Radford et al., 2021) model. NPR (Tan et al., 2024b) leverages distinctive upsampling artifacts inherent in generative models to develop a classifier based on pixel relationship patterns. DIRE (Wang et al., 2023), on the other hand, computes the diffusion inverse reconstruction error for both real and fake images and trains a detector to distinguish between these errors.

While current training-based detectors demonstrate promising results, they still have several limitations. First, their performance is heavily reliant on the quantity, quality, and diversity of the training data. Second, the training and re-training costs can be significant and scale unfavorably with the data volume. Finally, they often exhibit reduced generalization ability on images generated by new or unforeseen models. To circumvent these drawbacks, AEROBLADE (Ricker et al., 2024) presents a training-free solution by computing the

reconstruction error of a pretrained autoencoder only in the inference phase. Although AEROBLADE only shows good detection performance on images generated by LDM, it opens up new avenues for research in training-free AI-generated image detection.

In this paper, we aim to develop a more efficient training-free and generator-agnostic AI-generated image detection framework. We start by summarizing the lessons from existing studies as a unified paradigm: *the exploration of effective representations contrasting real versus AI-generated images is essential to successful detection.* This exploration has spanned various domains, including the frequency domain of images, the feature space of common classifiers, the representation space of pretrained large vision models, and the reconstruction error space. However, a crucial question remains: ***What kind of representation space is most suitable for detecting AI-generated images?***

Stein et al. (Stein et al., 2024) argue that models that consider both global image structure and key objects allow for a richer evaluation of a generative model. Motivated by this observation, we visualize the heatmap of different vision models by GradCAM (Selvaraju et al., 2017) on some images (upper left of Fig. 1). The results demonstrate that supervised models (ResNet 50 (He et al., 2016)) focus primarily on the main objects directly relevant to the classification result. In contrast, self-supervised models, particularly DINOv2 (Oquab et al., 2023), exhibit a more holistic perspective, capturing a broader understanding of the image content (Paul & Chen, 2022). Furthermore, we investigate the sensitivity of real and fake images to small perturbations, with a plot of the cosine similarity landscape (see Sec. 3.1 for details) shown in the upper right of Fig. 1. Our findings reveal that, compared to real images, AI-generated images exhibit higher sensitivity to small perturbations when using models like DINOv2, which adopts a more global view. Interestingly, this phenomenon is less obvious in ResNet 50 and CLIP. The reason could be that DINOv2 uses self-supervised learning on images only, while ResNet 50 uses image labels for supervised learning, and CLIP uses image captions for weakly supervised learning.

Taking advantage of this unique sensitivity property, we propose a **R**obust **AI**-**G**enerated **I**mage **D**etection method, **RIGID**. RIGID is a simple and efficient detection method. As shown in the bottom of Fig. 1, given an image, RIGID can effectively tell if it is real or AI-generated, by adding subtle noise and calculating the cosine similarity between the original and the noisy images to produce a detection score. Notably, RIGID **does not require any training or a priori knowledge of the generated images (e.g., which generator is used for generation)**. We evaluate the detection performance of RIGID on a wide range of AI-generated image datasets and benchmarks. The results show that RIGID, albeit a training-free method, can be competitive with extensively trained classifiers in several unseen and out-of-domain settings. Moreover, RIGID outperforms the state-of-the-art (SOTA) training-free method AEROBLADE by 26.07 and 28.49 AP points on the IMAGENET and LSUN-BEDROOM averages, respectively, rather than for every individual generator. A clear boundary case is DiT-XL/2: RIGID shows degraded performance on this highly photorealistic transformer-based diffusion generator, and we discuss this exception explicitly in Sec. 4 and Sec. 5. Furthermore, RIGID exhibits competitive performance across a broad range of generation techniques, robustness to common image corruptions, and strong generalization on unseen and out-of-domain scenarios (see Fig. 2).

We summarize our **main contributions** as follows:

- We propose RIGID, a simple training-free and generator-agnostic method for detecting AI-generated images. Under noise perturbation in the pixel space, RIGID leverages differentiated sensitivity in the representation space of a pretrained model to detect real versus AI-generated images.

- We interpret the perturbation score via Stein's lemma: it estimates the local sensitivity of a Gaussian-smoothed cosine similarity metric, as illustrated by Fig. 1 (top right panel). The local sensitivity is used as a feature to detect real or fake (AI-generated) images.

- Experiments show that RIGID is a lightweight complement to supervised and generative-prior detectors: it requires no fake training data, is efficient at inference time, and remains competitive across many evaluated generators while exposing DiT-XL/2 as an important boundary case.

- We further analyze threshold calibration, stochastic stability across noise seeds, runtime efficiency, and multi-perturbation averaging to clarify how RIGID can be used as a practical screening signal.

## 2 Related Works

**Image Generation** GANs and diffusion models are dominant image generation techniques. BigGAN (Brock et al., 2018) enhanced stability with orthogonal regularization, while StyleGAN (Karras et al., 2019) improved controllability using a style-based generator. DDPM (Ho et al., 2020) and LDM (Rombach et al., 2022a) achieve high-quality image generation. Conditional image generation, focusing on generating images from inputs like text, has seen advancements with GigaGAN (Kang et al., 2023) and ADM (Dhariwal & Nichol, 2021). DiT (Peebles & Xie, 2023) leverages Transformer's global context capture for improved text-to-image generation. These methods underpin popular tools like Stable Diffusion (Rombach et al., 2022b) and Midjourney (Midjourney, 2022).

**AI-generated Image Detection** Early AI-generated image detection relied on hand-crafted features like color (McCloskey & Albright, 2018; 2019) and co-occurrence features (Nataraj et al., 2019), but these became unreliable with advanced generative models. Frequency domain analysis (Frank et al., 2020; Dzanic et al., 2020; Chandrasegaran et al., 2021), while effective for upsampling models, fails to detect artifacts in diffusion model outputs (Corvi et al., 2023). Recent frequency-aware methods such as FreqNet (Tan et al., 2024a) revisit this direction by learning more general high-frequency cues. This signal is complementary to RIGID's representation-space sensitivity; in a preliminary study, combining RIGID's score with a simple frequency-band statistic improves DiT AP by approximately 6%, suggesting a useful future hybrid direction rather than a replacement for the core detector.

Training-based methods have shown promise. Training classifiers on GAN-generated images with augmentations (Gragnaniello et al., 2021) has yielded some generalization (Wang et al., 2020), while methods utilizing CLIP features (Ojha et al., 2023) or diffusion reconstruction errors (Wang et al., 2023) have also been explored. Recent CLIP-based detectors further show that pretrained vision-language features can be strong lightweight signals for synthetic-image detection (Cozzolino et al., 2024). Dataset-bias analyses also caution that JPEG compression and image-size artifacts can inflate detector performance if real and generated images are collected under mismatched preprocessing pipelines (Grommelt et al., 2024). However, these methods often suffer from limited generalizability and high computational costs. Training-free methods like AEROBLADE (Ricker et al., 2024), based on autoencoder reconstruction errors, offer an alternative solution. Nevertheless, it is only effective for images generated by LDM using similar autoencoders, and its generalizability remains a challenge. DiffusionFake (Sun et al., 2024) represents a generative-prior-based alternative that uses guided Stable Diffusion reconstruction to improve deepfake generalization; it is orthogonal to RIGID but heavier because it relies on a generative prior. Finally, recent adversarial studies show that AI-generated image detectors can be vulnerable to adaptive or black-box attacks (Diao et al., 2024), motivating the limitations discussion in Sec. 5.

## 3 Methodology

### 3.1 RIGID

**Design Objective** This work aims to develop an effective training-free method for detecting AI-generated images. Unlike existing training-free methods such as AEROBLADE (Ricker et al., 2024), which rely on the autoencoder used by LDM, our goal is to achieve effective detection across images produced by various generative methods without any prior knowledge of the generation process (i.e., a generator-agnostic detector). Notably, our approach does not change any component of the pretrained model, including the architecture and training weights. Detection solely uses the inference results of an off-the-shelf pretrained feature extractor to derive features differentiating real and generated images.

**Core Idea** While real and generated images often exhibit subtle differences in semantics and texture, these distinctions become increasingly difficult to discern by a human observer as generation methods advance. Current training-based detectors attempt to extract these hidden differences through supervised learning. Our work takes a different approach by exploiting the sensitivity difference of real and generated images to small perturbations. As shown in the upper right of Fig. 1, adding noise perturbations causes the features of real images to change continuously, resulting in a smoother gradient. Conversely, generated images are

more sensitive to noise, leading to a steeper change and gradient. Although the added noise is subtle, it can act as a probe for global features covering texture-rich and texture-poor regions of the image, which proves beneficial for generated image detection (Zhong et al., 2023). To accurately perceive how global features are affected by noise, we employ DINOv2 (Oquab et al., 2023) as our backbone model (feature extractor) since it has a holistic image view (Stein et al., 2024). A detailed discussion on the impact of different backbones on detection performance is provided in Sec. 4.4.

**Workflow** The workflow of RIGID is illustrated at the bottom of Fig. 1. Our proposed AI-generated image detector leverages the sensitivity difference between real and fake images to tiny perturbations for classification. Given an input sample, RIGID begins by adding subtle perturbations to the image. Then, both the original input sample and its noise-perturbed counterpart are fed into DINOv2 to obtain their feature embeddings. We separate the continuous similarity score from the binary deployment decision. The continuous score is

$$r(x) = \mathsf{sim}(f(x), f(x + \lambda \cdot \delta)); \qquad \delta \sim N(0, I), \tag{1}$$

where $f(\cdot)$ is the feature extractor, $\mathsf{sim}(\cdot)$ represents the cosine similarity between two embeddings, $\delta$ is the additive noise, and $\lambda$ is reserved for the RIGID perturbation intensity. Lower $r(x)$ indicates stronger perturbation sensitivity and therefore stronger evidence that the image is generated. For binary screening, we apply

$$S(x) = \mathbf{1}\{r(x) \le \epsilon\}. \tag{2}$$

The AUC and AP metrics reported in our experiments are computed from the continuous score and do not require selecting $\epsilon$. When a binary decision is needed, we calibrate $\epsilon$ on an independent held-out set of real images so that 95% of real images are accepted; this calibration uses no generated images. The expectation in Eq. 1 can also be approximated by averaging multiple independent perturbations,

$$r_K(x) = \frac{1}{K} \sum_{k=1}^{K} \mathsf{sim}(f(x), f(x + \lambda \cdot \delta_k)), \qquad \delta_k \sim N(0, I). \tag{3}$$

We keep $K = 1$ as the default because $K = 4$ and $K = 8$ provide only modest gains while increasing inference cost, as reported in Appendix E. Compared to existing methods, our approach offers several significant advantages:

- **Training-free:** RIGID operates solely during the inference phase, eliminating expensive training costs associated with methods such as (Corvi et al., 2023; Wang et al., 2020; Gragnaniello et al., 2021; Wang et al., 2023).

- **Generation-Independent:** Unlike AEROBLADE (Ricker et al., 2024), a training-free method that relies on an autoencoder closely tied to the underlying image generation model, RIGID utilizes DINOv2 (Oquab et al., 2023), a model trained with self-supervised learning without generated images.

- **Generator-agnostic:** RIGID does not assume knowledge of generation models, demonstrating the capability to detect a wide range of AI-generated images, including those from unseen generators and out-of-domain datasets (as summarized in Fig. 2).

- **Computationally Efficient:** Unlike DIRE (Wang et al., 2023) and AEROBLADE (Ricker et al., 2024), which need to compute reconstruction errors involving multi-step forward and backward diffusion processes via diffusion models, RIGID operates more efficiently by calculating embedding similarity directly.

### 3.2 Theoretical Analysis

The following analysis is intended as an interpretation of what the perturbation score measures, not as a proof that real and generated images must be separable. The method is motivated by the empirical sensitivity gap observed in self-supervised representation spaces; the role of Stein's lemma is to connect the score to

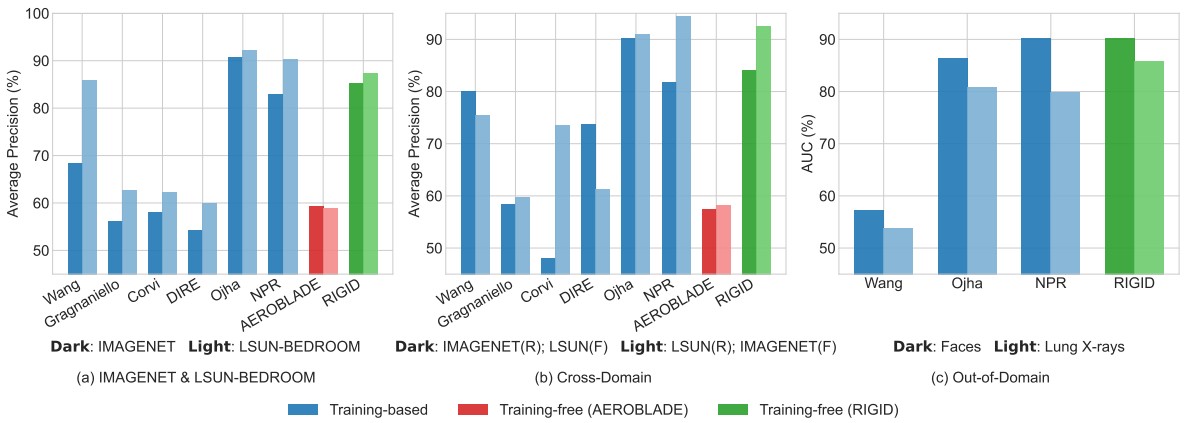

Figure 2: Comprehensive performance comparison of RIGID against baseline methods across multiple datasets and scenarios. Our training-free method RIGID is competitive with other training-free approaches and many training-based methods across: (a) Evaluations on IMAGENET and LSUN-BEDROOM. (b) Cross-domain evaluations (see Sec. 4.2.3), where R denotes real image and F denotes fake image. (c) Out-of-domain data (fully generated face detection and lung X-ray images). RIGID demonstrates strong generalization ability and robustness without requiring any training on AI-generated images.

local sensitivity. Given a backbone $f(\cdot) : \mathbb{R}^n \to \mathbb{R}^d$ and the cosine similarity function $h(\cdot) : \mathbb{R}^d \times \mathbb{R}^d \to \mathbb{R}$, the continuous score in Eq. 1 can be reformulated in expectation as:

$$G(x) = ((h \circ f) * N(0, \lambda^2 I))(x) = \mathbb{E}_{\delta \sim N(0, \lambda^2 I)}[h(f(x + \delta), f(x))] \tag{4}$$

where $*$ denotes the convolution operator between two functions, defined as $h * g = \int_{\mathbb{R}^d} h(t) g(x - t) dt$. Then, according to the Stein's lemma (Stein, 1981), $G(x)$ is differentiable with a gradient of:

$$
\begin{aligned}
\nabla G(x) &= \frac{1}{(2\pi\lambda^2)^{d/2}} \int_{\mathbb{R}^d} (h \circ f)(t) \frac{t - x}{\lambda^2} \exp\left(\frac{1}{2\lambda^2} \|x - t\|_2^2\right) dt \\
&= \frac{1}{\lambda^2} \mathbb{E}_{\delta \sim \mathcal{N}(0, \lambda^2 I)}[\delta \cdot h(f(x + \delta), f(x))]
\end{aligned}
\tag{5}
$$

Therefore, the random perturbation $\delta$ introduced by RIGID to $f(x + \delta)$ can be viewed as an operation of probing the gradient of the smoothed cosine similarity metric $G(x)$. According to the cosine similarity landscape in the upper right panel of Fig. 1, generated images empirically exhibit larger local sensitivity than real images under the DINOv2 representation. Here, Stein's lemma does not explain why this real/generated gap exists; that gap is established empirically through the score distributions and backbone ablations in Sec. 4.4. Thus, the analysis should be read as an interpretation that RIGID measures representation sensitivity through a smoothed similarity score.

## 4 Experiments

### 4.1 Experimental Setup

**Dataset** To comprehensively evaluate AI-generated image detectors, we use two rigorous test sets from (Stein et al., 2024). We assess detectors on IMAGENET (Deng et al., 2009) and LSUN-BEDROOM (Yu et al., 2015), using images generated by diverse SOTA models (Diffusion (Dhariwal & Nichol, 2021; Rombach et al., 2022a; Song et al., 2020; Ho et al., 2020; Wang et al., 2022; Peebles & Xie, 2023), GAN (Brock et al., 2018; Kang et al., 2023; Karras et al., 2019; Sauer et al., 2021; 2022), VAE (Lee et al., 2022), Transformer (Bond-Taylor et al., 2022), Mask Prediction (Chang et al., 2022)) selected from a leaderboard (with Code), ensuring representation of cutting-edge generative capabilities.

Table 1: The AUC and AP of different AI-generated image detectors on IMAGENET. A higher value indicates better performance. The **bolded** values are the best performance, and the _underlined italicized_ values are the second-best performance. The same annotation holds for all tables.

| AUC/AP (%) | Training Samples | Diffusion | | | | GAN | | | | VAE | Average |
|---|---|---|---|---|---|---|---|---|---|---|---|
| | | ADM | ADMG | LDM | DiT | BigGAN | GigaGAN | StyleGAN XL | RQ-Transformer | Mask GIT | |
| Wang | 720 000 | 65.96/66.75 | 65.56/66.59 | 67.82/69.43 | 61.97/64.25 | 83.15/84.76 | 71.19/69.96 | 66.63/66.06 | 60.66/61.67 | 65.43/66.97 | 67.60/68.43 |
| Gragnaniello | 400 000 | 60.21/59.91 | 59.45/59.71 | 61.61/61.37 | 56.67/56.56 | 59.62/58.49 | 53.63/52.35 | 51.58/52.35 | 56.49/54.34 | 53.70/52.68 | 56.99/56.24 |
| Corvi | 400 000 | 63.94/63.85 | 65.55/65.19 | 62.18/60.83 | 56.64/55.23 | 61.91/59.95 | 50.15/49.18 | 48.48/48.05 | 63.21/60.48 | 61.19/59.51 | 59.25/58.03 |
| DIRE | 80 000 | 57.79/56.67 | 57.09/56.80 | 61.47/62.15 | 53.21/53.52 | 49.63/50.00 | 50.00/51.14 | 52.91/53.87 | 53.17/52.41 | 49.93/51.57 | 53.91/54.24 |
| Ojha | 720 000 | **90.90/90.76** | **87.13/87.20** | _86.55/86.36_ | _81.67/81.86_ | **96.31/96.24** | **93.54/93.55** | **92.16/92.13** | **94.12/93.79** | **95.28/95.05** | **90.85/90.77** |
| NPR | 720 000 | 83.73/81.26 | _84.01/83.14_ | **94.43/91.36** | **83.12/82.78** | 89.85/87.42 | 79.70/78.56 | 77.88/75.83 | 77.31/75.19 | 92.03/90.73 | 84.67/82.92 |
| AEROBLADE | Training Free | 52.20/53.65 | 59.24/57.93 | 62.97/61.96 | 72.98/73.65 | 50.07/50.94 | 55.21/54.87 | 51.17/52.85 | 70.23/69.36 | 59.80/58.71 | 59.32/59.33 |
| RIGID | Training Free | _87.75/86.06_ | 83.50/81.46 | 81.50/80.23 | 72.07/69.55 | _93.86/93.57_ | _89.29/87.92_ | _85.94/84.75_ | _93.39/93.11_ | _92.65/91.91_ | _86.67/85.40_ |

Table 2: The AUC and AP of different AI-generated image detectors on LSUN-BEDROOM.

| AUC/AP (%) | Training Samples | ADM | DDPM | iDDPM | Diffusion Projected GAN | Projected GAN | StyleGAN | Unleashing Transformer | Average |
|---|---|---|---|---|---|---|---|---|---|
| Wang | 720 000 | 66.13/65.96 | 81.87/82.07 | 78.46/79.13 | 90.63/90.59 | 92.55/92.43 | **98.47/98.34** | 92.55/92.66 | 85.81/85.88 |
| Gragnaniello | 400 000 | 55.92/57.46 | 65.58/65.99 | 62.47/62.87 | 59.15/57.95 | 63.36/62.36 | 67.08/66.01 | 66.12/67.00 | 62.96/62.81 |
| Corvi | 400 000 | 56.67/58.21 | 68.67/70.02 | 68.70/69.57 | 55.46/54.94 | 54.54/55.16 | 54.26/55.71 | 72.44/71.91 | 61.54/62.22 |
| DIRE | 80 000 | 56.36/57.26 | 60.29/60.87 | 63.52/63.74 | 56.31/55.89 | 57.42/58.14 | 58.38/58.83 | 64.77/65.26 | 59.58/60.00 |
| Ojha | 720 000 | _82.37/82.66_ | _90.88/90.66_ | _91.92/_**92.02** | _95.02/_94.85 | **96.73/96.63** | 91.92/_91.88_ | _96.94/_**96.84** | _92.25/_**92.22** |
| NPR | 720 000 | **85.05/82.73** | **96.58/92.42** | **92.29/**_90.70_ | **95.51/**92.35 | 93.61/90.72 | _92.66/_89.88 | **97.80/**_94.67_ | **93.36/**_90.41_ |
| AEROBLADE | Training Free | 58.03/59.33 | 73.92/74.31 | 68.20/69.18 | 51.46/50.00 | 52.10/50.81 | 52.60/50.81 | 61.19/58.34 | 59.46/58.98 |
| RIGID | Training Free | 74.04/72.92 | 89.30/89.76 | 85.61/86.07 | 93.86/_94.49_ | _94.41/94.81_ | 84.12/81.53 | 92.49/92.63 | 87.69/87.47 |

Furthermore, we expand evaluation to images from popular platforms like Stable Diffusion (Rombach et al., 2022b), Midjourney (Midjourney, 2022), and Wukong (Gu et al., 2022), sourced from the GenImage (Zhu et al., 2024) benchmark. We also evaluate on out-of-distribution datasets including fully generated face images (140k Real & Faces, 2020) and lung X-ray (Ali et al., 2022) images to further assess generalization capabilities. This diverse range of generative models and datasets allows for a more robust and generalizable assessment of detector performance. A detailed description of the datasets used in our evaluation can be found in the Appendix.

**Evaluation Metrics** Following existing detection methods (Corvi et al., 2023; Wang et al., 2020), we primarily utilize two key metrics to evaluate the performance of the detectors in our experiments: Area Under the Receiver Operating Characteristic curve (AUC) and Average Precision (AP). Both AUC and AP provide a quantitative measure of detection accuracy, with higher scores indicating better performance. Both metrics are threshold-independent and are computed from the continuous RIGID score rather than from the binary decision threshold $\epsilon$. For deployment-oriented binary screening, Appendix E reports thresholds calibrated to accept 95% of held-out real images and the corresponding thresholded accuracy.

**Baselines** We conduct a comparative analysis of RIGID against a range of established AI-generated image detection methods, encompassing both training-based and training-free approaches. The former include Wang et al (Wang et al., 2020), Gragnaniello et al (Gragnaniello et al., 2021), Corvi et al (Corvi et al., 2023), DIRE (Wang et al., 2023), Ojha (Ojha et al., 2023) and NPR (Tan et al., 2024b). The latter includes a prominent training-free method: AEROBLADE (Ricker et al., 2024). Detailed information regarding the implementation of these baseline methods can be found in the Appendix.

## 4.2 Evaluation of Detection Performance

Fig. 2 provides a comprehensive overview of RIGID's performance compared to baseline methods across multiple datasets and evaluation scenarios. As shown, RIGID is competitive with other training-free approaches and with many training-based methods, especially in unseen or out-of-domain settings. This overview highlights three key aspects of our evaluation: (a) Evaluations on standard datasets (IMAGENET and LSUN-BEDROOM), (b) Cross-domain performance where training and testing distributions differ, and (c) out-of-domain datasets such as fully generated face detection and medical images. The consistent performance across these diverse scenarios underscores RIGID's robustness and generalizability.

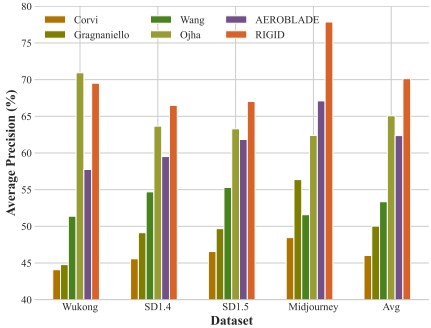
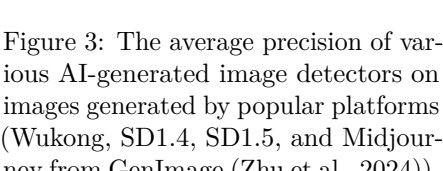

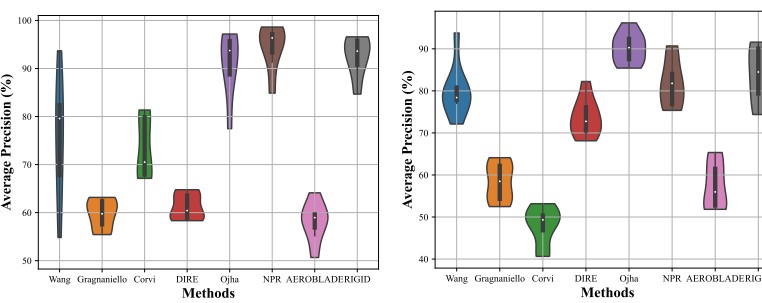

(a) R: IMAGENET; F: LSUN-BEDROOM    (b) R: LSUN-BEDROOM; F: IMAGENET

Figure 3: The average precision of various AI-generated image detectors on images generated by popular platforms (Wukong, SD1.4, SD1.5, and Midjourney from GenImage (Zhu et al., 2024)).

Figure 4: **Cross-dataset Evaluation** on IMAGENET and LSUN-BEDROOM. The violin graph shows AP distribution, where the black bar in the center indicates the interquartile range and the white dot is the median. R: Real images, F: Fake images.

### 4.2.1 Testing on ImageNet and LSUN-Bedroom

We comprehensively evaluate AI-generated image detection methods on IMAGENET and LSUN-BEDROOM (Tables 1 and 2). Note that DIRE's performance is lower than reported due to format bias (Ricker et al., 2024). Our analysis reveals several key findings of RIGID:

**1) Superior Performance.** RIGID improves over AEROBLADE by more than 25% AP points on the IMAGENET and LSUN-BEDROOM average rows, as summarized in Appendix Table 4. This comparison is limited to these evaluated averages and should not be read as a per-generator guarantee. DiT-XL/2 is the clearest exception: on IMAGENET, RIGID obtains 69.55% AP while AEROBLADE obtains 73.65% AP, and trained detectors such as Ojha and NPR also perform better on this generator.

**2) Strong Generalization Ability.** RIGID effectively detects images from diverse generation methods across both datasets. Its performance should be understood as competitive across a broad generator set rather than uniformly dominant. In contrast, training-based methods show significant limitations: Wang et al.'s method (trained on ProGAN) performs poorly on diffusion models. NPR (trained on LSUN) shows performance degradation on IMAGENET, with average performance inferior to RIGID.

**3) Independence from Generation Bias.** Unlike AEROBLADE, which depends on autoencoders from generative models, RIGID operates independently using only DINOv2, a self-supervised vision model. AEROBLADE's performance varies significantly based on whether test images were generated using autoencoders similar to its own, which is evident in its improved performance on images generated by methods using autoencoders (LDM, DiT, RQ-Transformer). For DiT-XL/2, we observe that the cosine-similarity gap $\Delta = \mathsf{sim}_{real} - \mathsf{sim}_{fake}$ shrinks to approximately 0.015, compared with roughly 0.04–0.08 for most other generators. This reduced margin is consistent with DiT's latent-space smoothness and its globally coherent samples, which contain fewer high-frequency artifacts that RIGID partly relies on. AEROBLADE benefits in this setting because DiT shares a related VAE family, while Ojha benefits from CLIP's broad supervised training distribution; neither advantage directly transfers to RIGID's training-free sensitivity score.

### 4.2.2 Evaluation on Popular Text-to-Image Generation Platforms

Fig. 3 compares the detection performance of RIGID and other detection methods on images generated by four widely used platforms: Wukong, SD 1.4, SD 1.5 and Midjourney. All images are extracted from the GenImage benchmark (Zhu et al., 2024). In this setting, training-free methods outperform training-based methods such as Ojha, likely because the training data lags behind rapidly evolving generation techniques. This highlights the need for effective, stable, and training-free detection. On this GenImage platform evaluation, RIGID achieves the highest average AP among the compared methods, supporting its usefulness under fast generator turnover while not implying universal superiority over trained detectors.

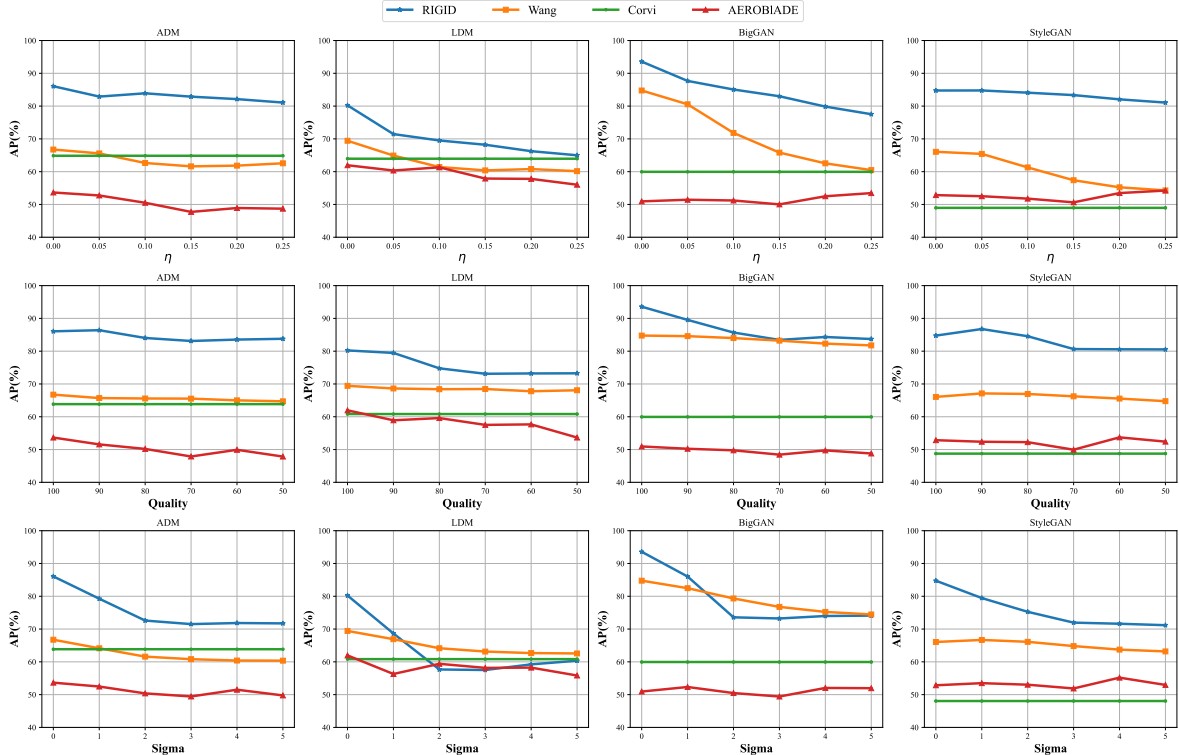

Figure 5: **Robustness to Image Corruptions.** The top row shows the robustness to Gaussian noise ($\eta$ represents the corruption severity). The second row shows the robustness to JPEG compression, and the bottom row shows the robustness to Gaussian blur.

### 4.2.3 Cross Domain and Out-of-domain Testing

Following (Wang et al., 2023), we evaluate detection methods under domain shifts where real and fake distributions differ. Fig. 4 shows two cross-domain scenarios: (a) real images from IMAGENET with fake images from LSUN-BEDROOM, and (b) the reverse configuration. RIGID maintains stable performance across both scenarios, demonstrating robust domain shift resilience. In contrast, training-based methods show significant AP decline when tested on distributions different from their training data. RIGID even outperforms Ojha in Fig. 4 (a) and NPR in Fig. 4 (b).

For extreme generalization testing, Fig. 2 (c) shows results on entirely different domains: fully generated face images (140k Real & Faces, 2020) and medical X-rays (Ali et al., 2022). Even in these specialized visual domains, RIGID maintains strong detection performance ($> 80\%$ AUC), significantly outperforming baselines. We also report FaceForensics++ DeepFake Subset results in Appendix Table 5; these results support generalization across different face-generation settings, while face manipulation detection is not the primary focus of this work. This exceptional out-of-domain capability further confirms RIGID's generator-agnostic design enables effective deployment across diverse scenarios.

### 4.3 Robustness to Image Corruptions

Following (Wang et al., 2023; Ricker et al., 2024), we evaluate detector robustness against three common image corruptions: Gaussian noise, JPEG compression, and Gaussian blur. As shown in Fig. 5, we test each corruption at five intensity levels ($\eta = \{0.05, 0.1, 0.15, 0.2, 0.25\}$; Quality= $\{90, 80, 70, 60, 50\}$; Sigma= $\{1, 2, 3, 4, 5\}$) across four generation methods: ADM (Dhariwal & Nichol, 2021), LDM (Rombach et al., 2022a), BigGAN (Brock et al., 2018), and StyleGAN (Karras et al., 2019).

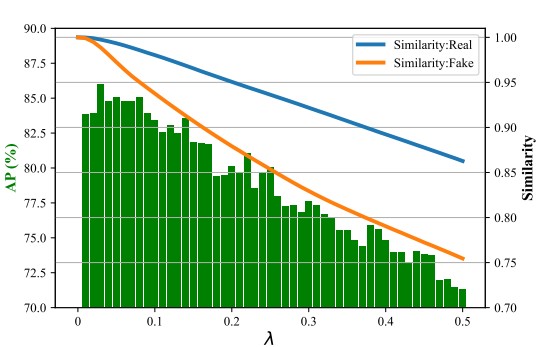

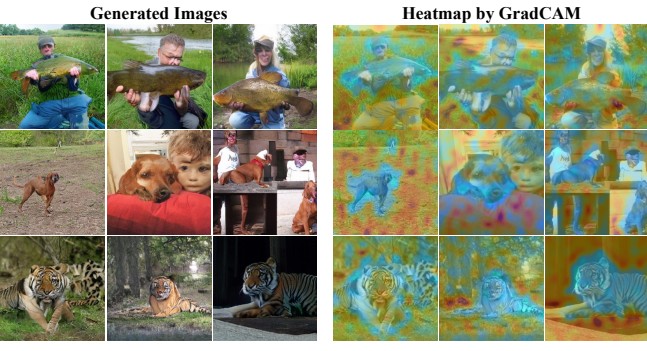

Figure 6: **Detection performance for different noise intensities (the value $\lambda$ in eq. 2).** The left/right y-axis is AP/Cosine-Similarity.

Figure 7: **Interpretability visualization of AI-generated image detection using RIGID.** Note that higher/lower heat levels represent areas identified as real/AI-generated by GradCAM. Some low-quality images are deliberately selected here solely for demonstration.

RIGID consistently outperforms baseline methods under most corruption conditions, demonstrating superior resilience. It maintains a significant advantage over AEROBLADE across all corruption types and generation models. While training-based methods show less degradation under JPEG compression and Gaussian blur (likely due to these corruptions being included in their training augmentations), they perform poorly with unfamiliar corruptions like Gaussian noise. For example, Wang et al.'s method experiences only a 3% performance drop with JPEG compression but a substantial 13% drop with Gaussian noise.

These results highlight RIGID's intrinsic robustness to image corruptions without requiring specific training accommodations, demonstrating its reliability even when processing degraded images.

### 4.4 Ablation Studies

**Noise Intensity** Fig. 6 illustrates the impact of noise intensity ($\lambda$) on RIGID's performance, alongside the trend of cosine similarity between real and generated (fake) images. At $\lambda = 0$, both real and generated images exhibit a cosine similarity of 1, resulting in an AP of approximately 50%, equivalent to random guessing. As noise intensity increases, the disparity in cosine similarity between real and generated images widens. However, excessively high noise levels negatively impact RIGID's detection performance, likely due to the disruption of normal feature representation caused by the noise. Within a moderate noise range (0 to 0.17), RIGID maintains high detection performance with AP scores greater than or equal to 80%. Importantly, even under very high noise levels, RIGID continues to outperform the baseline methods listed in Table 1. This demonstrates that RIGID is not a hyperparameter-sensitive method. We also evaluate whether RIGID depends on a special perturbation distribution. Table 3 shows that Gaussian, Gamma, Laplace, and Chi-square perturbations yield similar IMAGENET average AP values in the 84.55%–85.40% range, indicating that the method is driven by representation sensitivity rather than by one particular noise family. Because RIGID samples random perturbations, we quantify stochastic stability using five independent noise seeds in Appendix Table 7. The standard deviations are small on both IMAGENET and LSUN-BEDROOM, confirming that the detection signal is stable across noise initializations.

**Backbone** Fig. 8 provides visual comparisons of the interest regions identified by different backbones in RIGID and their corresponding performance in detecting AI-generated images. The heatmaps on the left of Fig. 8 reveal distinct patterns in how each backbone perceives image features: ResNet50 and CLIP exhibit a more localized focus, highlighting specific regions within the images. SAM (Kirillov et al., 2023) and DINOv2 show a more balanced focus, capturing both local details and global context. The boxplot on the right of Fig. 8 compares the Average Precision of each backbone in detecting generated images. Notably, SAM and DINOv2, with their holistic approach to image understanding, achieve significantly higher AP scores than

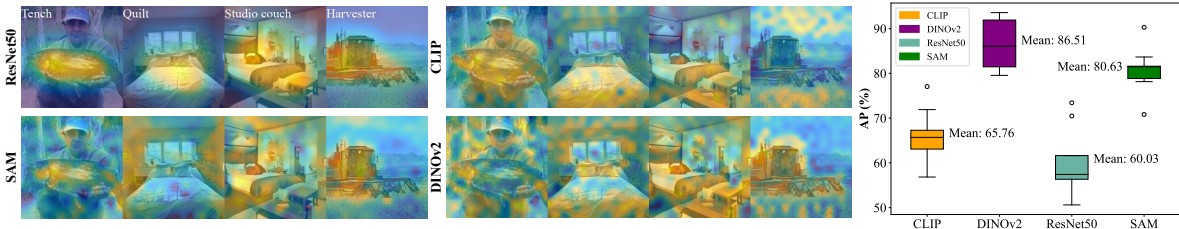

Figure 8: **Detection performance using different backbones.** The heatmap on the left visualizes what the Fréchet Distance (Heusel et al., 2017) perceives for each backbone. The right part shows the detection performance using different backbones.

Table 3: The AP of noise from different distributions on IMAGENET. A higher value indicates better performance.

| Distribution | ADM | ADMG | LDM | DiT | BigGAN | GigaGAN | StyleGAN XL | RQ-Transformer | Mask GIT | Aver |
|---|---|---|---|---|---|---|---|---|---|---|
| Laplace | 86.36 | 79.49 | 78.57 | 67.91 | 93.98 | 86.49 | 84.53 | 92.65 | 90.94 | 84.55 |
| Gamma | 85.96 | 80.51 | 78.58 | 71.82 | 93.15 | 88.70 | 84.73 | 93.24 | 90.82 | 85.28 |
| Chi-Square | 86.65 | 79.74 | 75.86 | 68.09 | 94.76 | 88.25 | 86.42 | 92.73 | 91.45 | 84.88 |
| Gaussian | 86.06 | 81.46 | 80.23 | 69.55 | 93.57 | 87.92 | 84.75 | 93.11 | 91.91 | 85.40 |

locally-focused backbones (ResNet50 and CLIP), underscoring the importance of a holistic view for effective AI-generated image detection. This insight informed RIGID's backbone choice.

### 4.5 Interpretability Analysis

To demonstrate that RIGID's detection mechanism specifically targets generation artifacts, we employ GradCAM (Selvaraju et al., 2017) visualization on deliberately selected samples exhibiting poor generation quality with artifacts perceptible to average observers. These lower-quality samples are used to illustrate artifact-dense regions, not to claim that GradCAM explains the full theoretical mechanism. In Fig. 7, the heatmap intensity reflects cosine similarity: high-heat areas (red) denote regions maintaining similarity under perturbation, indicating authentic characteristics, whereas low-heat areas (blue) signify sensitivity to noise, exposing AI-generated content. The visualization provides qualitative evidence that RIGID is perceptually sensitive to artifact-rich regions, including texture discontinuities, geometric inconsistencies, and edge anomalies. It does not by itself explain why the representation space has this sensitivity; that question remains an empirical and theoretical topic for future work.

### 4.6 Efficiency and Additional Analyses

Appendix Table 9 reports runtime measurements on the same hardware. RIGID requires 11.8 ms/image and reaches 84.7 images/s, making it faster than KNN, VIM, AEROBLADE, and DIRE in our measurement. This supports its intended use as a lightweight screening signal before heavier detectors or manual inspection. Appendix Table 8 reports the effect of averaging multiple perturbations. Increasing from $K = 1$ to $K = 4$ or $K = 8$ gives modest AUC improvements on IMAGENET and LSUN-BEDROOM, but the gains are small relative to the added computation; therefore, we keep $K = 1$ as the default.

## 5 Discussion

**Limitations of training-based methods:** While training-based AI-generated image detectors (Corvi et al., 2023; Wang et al., 2020; Gragnaniello et al., 2021; Wang et al., 2023) can perform well in controlled settings, they face significant limitations: (a) **Expensive training cost.** Effective detectors require substantial computational resources and extensive data collection. (b) **Dependence on training data quality and**

**quantity.** Tables 1 and 2 show that detectors with more training samples achieve higher average performance, but acquiring diverse, high-quality generated images remains costly. (c) **Hyperparameter sensitivity.** Fine-tuning numerous hyperparameters (augmentation methods, learning rates, etc.) further increases the computational cost. (d) **Poor generalization.** Fig. 2 clearly shows that the training-based detector generalizes poorly to generation styles different from the training data. This does not mean that RIGID is universally superior to trained detectors. Ojha, NPR, and related classifiers remain preferable when representative fake data, retraining resources, and stable deployment domains are available. RIGID is useful in a different operational niche: no target fake data, fast generator turnover, only real validation data for calibration, or lightweight triage before heavier analysis.

**Limitations of training-free methods:** While addressing training costs and improving generalization, training-free approaches also have limitations: (a) **Reliance on pretrained models.** These detectors may inherit biases from their foundation models. AEROBLADE's dependence on LDM autoencoders significantly limits its effectiveness on images from different generative architectures. (b) **Performance degradation on high-quality generated images.** As shown in Table 1, training-free methods (both AEROBLADE and RIGID) struggle to achieve high detection accuracy on high-quality generated images (e.g., DiT-XL2). For RIGID, the DiT-XL/2 sensitivity gap is much smaller than for most other evaluated generators, suggesting that highly photorealistic transformer-based diffusion generators remain an open challenge. Several additional limitations are important for deployment. First, thresholds calibrated on one real-image distribution may transfer imperfectly to another, which can increase false positives or false negatives. Second, fully generated face detection is evaluated here, but face manipulation detection is not the main focus and may require specialized evidence. Third, RIGID uses a public frozen backbone and a scalar threshold, so adaptive adversaries may optimize images to preserve DINOv2 similarity after perturbation. We therefore position RIGID as an effective AI-generated image detector and screening signal, not as a forensic system with adaptive adversarial robustness guarantees.

**Broader Impact.** RIGID can help flag AI-generated images in settings where fake samples from new generators are unavailable, but its outputs should be treated as probabilistic screening evidence rather than final forensic proof. Practical use should account for threshold transfer, high-fidelity generator failures, possible false positives on unusual real images, false negatives on face manipulations or DiT-like generators, and adaptive evasion. Transparent reporting of calibration data and uncertainty is therefore important when deploying the detector.

# 6 Conclusion

This paper introduced RIGID, a novel training-free and generator-agnostic method for robust detection of AI-generated images. Based on our key observation that real images exhibit less sensitivity to random perturbations in the representation space, RIGID effectively uses this property to distinguish between real and AI-generated images by comparing the representation similarity before and after noise perturbation. Our extensive evaluations demonstrate that RIGID is a lightweight complement to supervised and generative-prior detectors, with strong average performance on the evaluated benchmarks and useful generalization to unseen generation methods and out-of-domain data. This generalization capability, coupled with RIGID's resilience to diverse image corruptions, establishes it as a practical screening signal for detecting AI-generated images.

### Acknowledgments

Zhiyuan He and Tsung-Yi Ho, from the JC STEM Lab of Intelligent Design Automation, are funded by the Hong Kong Jockey Club Charities Trust. Pin-Yu Chen is fully funded by IBM Research.

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

## A  Experimental Details

All our experiments were tested on a NVIDIA GeForce RTX 3090 with 24G memory. The model we used is DINOv2 (Oquab et al., 2023) VIT Large with a patch size of 14, and the noise intensity $\lambda$ is 0.05.

## B  Cosine Similarity Landscape

Following (Li et al., 2018), we plot the cosine similarity landscape of real and generated images. The plot function is defined as follows:

$$f(x|\alpha, \beta) = \frac{1}{|X|} \sum_{x \in X} \mathsf{sim}[f_\theta(x \oplus (\alpha \mathbf{u} + \beta \mathbf{v})), f_\theta(x)] \tag{6}$$

Where $X$ represents the sample set of real images or generated images, $sim$ is the cosine similarity, $f_\theta(\cdot)$ is a feature extractor, and $\mathbf{u}$ and $\mathbf{v}$ are two random direction vectors sampled from the Gaussian distribution. We plot the cosine similarity landscape of ResNet50, CLIP and DINOv2 in Fig. 1 in the main paper. In our experiments, $\alpha$ and $\beta$ range from -0.5 to 0.5 with a step size of 0.01.

## C  Generated Datasets

The generated images on IMAGENET and LSUN-BEDROOM we used are both from (Stein et al., 2024), which generated 100,000 images for each generation model in each dataset based on the leaderboard (with Code) of generation quality on the two datasets. For class-conditional models, the same number of samples from each class is generated, i.e. 100 images per class in IMAGENET. The repository link and FID scores of different generation methods on IMAGENET and LSUN-BEDROOM are as follows:

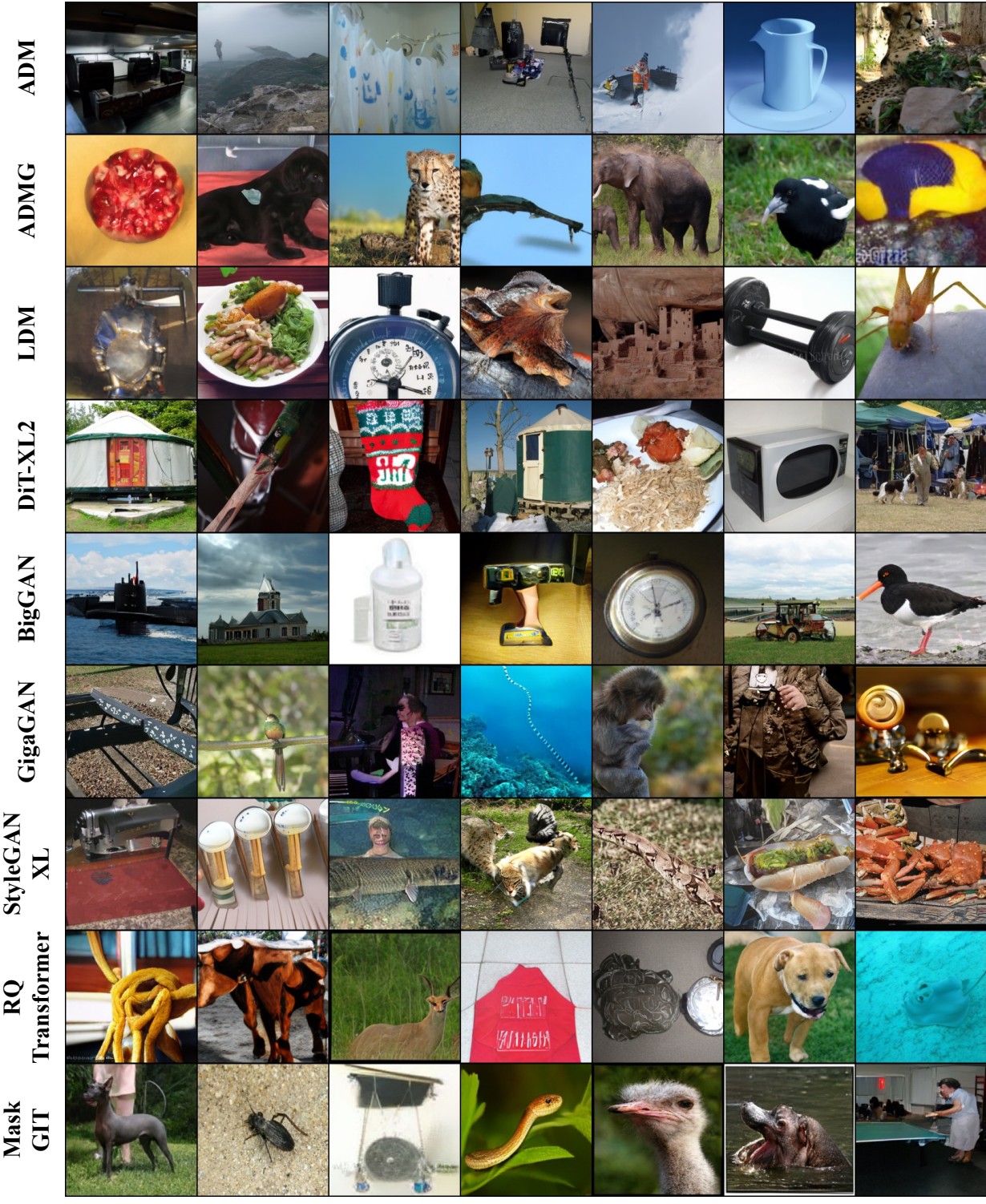

Figure 9: **Display of Generated Images on ImageNet.** Generation methods include: ADM, ADMG, LDM, DiT-XL2, BigGAN, GigaGAN, StyleGAN-XL, RQ-Transformer and MaskGIT.

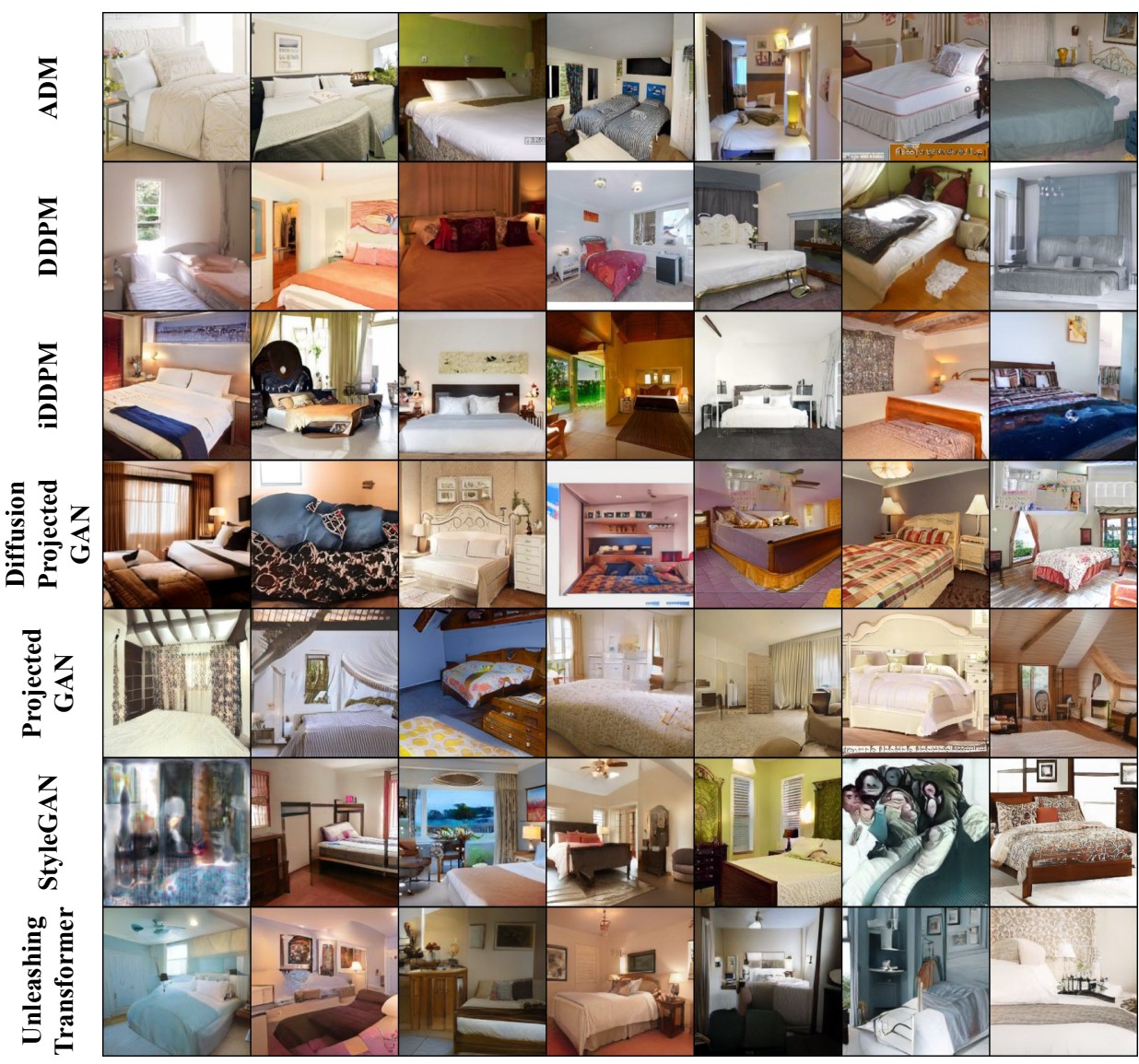

Figure 10: **Display of Generated Images on LSUN-Bedroom.** Generation methods include: ADM, DDPM, iDDPM, Diffusion Projected GAN, Projected GAN, StyleGAN and Unleashing Transformer.

### C.1 ImageNet

- Three models used sets of 50k publicly available images provided at `https://github.com/openai/guided-diffusion/tree/main/evaluations`

  - **ADM** (Dhariwal & Nichol, 2021). FID=11.84
  - **ADMG** (Dhariwal & Nichol, 2021). FID=5.58
  - **BigGAN** (Brock et al., 2018). FID=7.94

- **DiT-XL-2** (Peebles & Xie, 2023). FID=2.80. `https://github.com/facebookresearch/DiT`.

- **GigaGAN** (Kang et al., 2023). With 100k images provided privately by authors. FID=4.16.

- **LDM** (Rombach et al., 2022a). FID=4.29. `https://github.com/CompVis/latent-diffusion`.

- **StyleGAN-XL** (Sauer et al., 2022). FID=2.91. `https://github.com/autonomousvision/stylegan-xl`.

- **RQ-Transformer** (Lee et al., 2022). FID=9.71. `https://github.com/kakaobrain/rq-vae-transformer`.

- **Mask-GIT** (Chang et al., 2022). FID=5.63. `https://github.com/google-research/maskgit`.

## C.2 LSUN-Bedroom

- Three models used sets of 50k publicly available images provided at `https://github.com/openai/guided-diffusion/tree/main/evaluations`.

  - **ADM** (Dhariwal & Nichol, 2021). FID=2.20
  - **DDPM** (Ho et al., 2020). FID=5.18.
  - **iDDPM** (Nichol & Dhariwal, 2021). FID=4.54.
  - **StyleGAN** (Karras et al., 2019). FID=2.65.

- **Diffusion-Projected GAN** (Wang et al., 2022). FID=1.79. `https://github.com/Zhendong-Wang/Diffusion-GAN`.

- **Projected GAN** (Sauer et al., 2021). FID=2.23. `https://github.com/autonomousvision/projected-gan`.

- **Unleashing Transformers** (Bond-Taylor et al., 2022). FID=3.58. `https://github.com/samb-t/unleashing-transformers`.

## C.3 GenImage

GenImage (Zhu et al., 2024) is the latest million-level benchmark for detecting AI-generated images. One of the advantages of GenImage is that it contains generated images from four mainstream text-to-image platforms, including: Wukong (Gu et al., 2022), SD 1.4 (Rombach et al., 2022b), SD 1.5 (Rombach et al., 2022b) and Midjourney (Midjourney, 2022). GenImage input sentences follow the template "photo of class", where "class" is replaced by ImageNet labels. For Wukong, Chinese sentences tend to achieve better generation quality. In this way, the sentences are translated into Chinese in advance.

## C.4 Out-of-Domain Images

- **140k Real and Fake Faces.** `https://www.kaggle.com/datasets/xhlulu/140k-real-and-fake-faces`

- **FaceForensics++ DeepFake Subset.** We use this subset as an additional face-domain evaluation to test generalization beyond fully generated StyleGAN-style faces.

- **Lung X-ray Images** `https://www.kaggle.com/datasets/hazrat/awesomelungs`

# D Baselines

**Wang et al.** (Wang et al., 2020) We use the code and model checkpoints from the official repository[1].

**Gragnaniello et al.** (Gragnaniello et al., 2021) and **Corvi et al.** (Corvi et al., 2023) we use the code and model checkpoints from the official repository[2] provided by Corvi et al. This repository also includes the detector from Gragnaniello et al.

---

[1]https://github.com/PeterWang512/CNNDetection
[2]https://github.com/grip-unina/DMimageDetection

**DIRE** (Wang et al., 2023) We use the code and model checkpoints from the official repository[3]. However, (Ricker et al., 2024) points out that the excellent performance reported in DIRE comes from saving real images as JPEG files and generated images as PNG files, which causes DIRE to learn the differences between formats. Therefore, we converted both real images and generated images into JPEG format and tested their performance as shown in Tables 1 and 2 in the main paper.

**Ojha** We use the code and model checkpoints from the official repository[4].

**NPR** We use the code and model checkpoints from the official repository[5].

**AEROBLADE** (Ricker et al., 2024) We use the code from the official repository[6]. We use the autoencoder from CompVis-stable-diffusion-v1-1-ViT-L-14-openai to compute the reconstruction error.

# E  Additional Results

Table 4 summarizes the AEROBLADE comparison on the evaluated dataset averages.

Table 4: AEROBLADE comparison on evaluated dataset averages. The advantage holds on these evaluated averages, not for every individual generator; DiT-XL/2 is an exception discussed in the main text.

| Setting | AEROBLADE AP | RIGID AP | Difference |
|---|---|---|---|
| IMAGENET avg. | 59.33 | 85.40 | +26.07 |
| LSUN-BEDROOM avg. | 58.98 | 87.47 | +28.49 |

Table 5 reports the additional face-domain evaluation, including both fully generated faces and FaceForensics++ DeepFake samples.

Table 5: Additional face-domain evaluation. The first row evaluates fully generated faces, while the second row evaluates the FaceForensics++ DeepFake Subset.

| Dataset | AUC | AP |
|---|---|---|
| 140k Real/Fake Faces | 90.14 | 89.92 |
| FaceForensics++ avg. | 87.32 | 86.58 |

Table 6 provides deployment-oriented threshold calibration results using held-out real images only.

Table 6: Threshold calibration for binary screening. Each $\epsilon$ is calibrated on held-out real images to accept 95% of real images; AUC and AP in the main experiments remain threshold-independent.

| Target | $\epsilon$ | Acc |
|---|---|---|
| IMAGENET | 0.924 | 83.21 |
| LSUN-BEDROOM | 0.919 | 84.48 |
| Faces | 0.913 | 88.35 |
| X-ray | 0.909 | 82.67 |

Table 7 quantifies the stochastic stability of RIGID under different random noise seeds.

Table 8 evaluates multi-perturbation averaging and shows the diminishing returns of increasing $K$.

Table 9 compares inference efficiency and highlights RIGID's lightweight runtime profile.

---

[3]https://github.com/ZhendongWang6/DIRE
[4]https://github.com/WisconsinAIVision/UniversalFakeDetect
[5]https://github.com/chuangchuangtan/NPR-DeepfakeDetection
[6]https://github.com/jonasricker/aeroblade

Table 7: Stability over five independent random noise seeds. The small standard deviations indicate that RIGID is not strongly dependent on any specific noise instance.

| Setting | AUC mean $\pm$ std | AP mean $\pm$ std |
|---|---|---|
| ImageNet avg. | 86.58 $\pm$ 0.18 | 84.96 $\pm$ 0.24 |
| LSUN-Bedroom avg. | 87.82 $\pm$ 0.21 | 87.51 $\pm$ 0.27 |

Table 8: Effect of averaging multiple perturbed versions. Averaging improves AUC, but the gains diminish as $K$ increases, so we keep $K = 1$ as the default.

| Setting | $K = 1$ AUC | $K = 4$ AUC | $K = 8$ AUC |
|---|---|---|---|
| ImageNet average | 86.67 | 86.94 | 87.19 |
| LSUN-Bedroom average | 87.69 | 88.03 | 88.27 |

## F  Display of Generated Images

We display images generated by different generation methods on ImageNet and LSUN-Bedroom in Fig. 9 and Fig. 10.

## G  Display of Perturbed Images

We display images perturbed by three different perturbation methods: Gaussian Noise, JPEG Compression and Gaussian Blur in Fig. 11. For each perturbation, we set five levels, including $\eta = 0.05, 0.1, 0.15, 0.2, 0.25$, $q = 90, 80, 70, 60, 50$ and $\gamma = 1.0, 2.0, 3.0, 4.0, 5.0$.

Table 9: Runtime comparison. RIGID achieves the highest throughput in this measurement while requiring only backbone forward passes and a similarity computation.

| Method | ms/image | images/s |
|---|---|---|
| RIGID | 11.8 | 84.7 |
| KNN | 32.5 | 30.7 |
| VIM | 16.3 | 61.3 |
| AEROBLADE | 93.5 | 10.7 |
| DIRE | 218.6 | 4.6 |

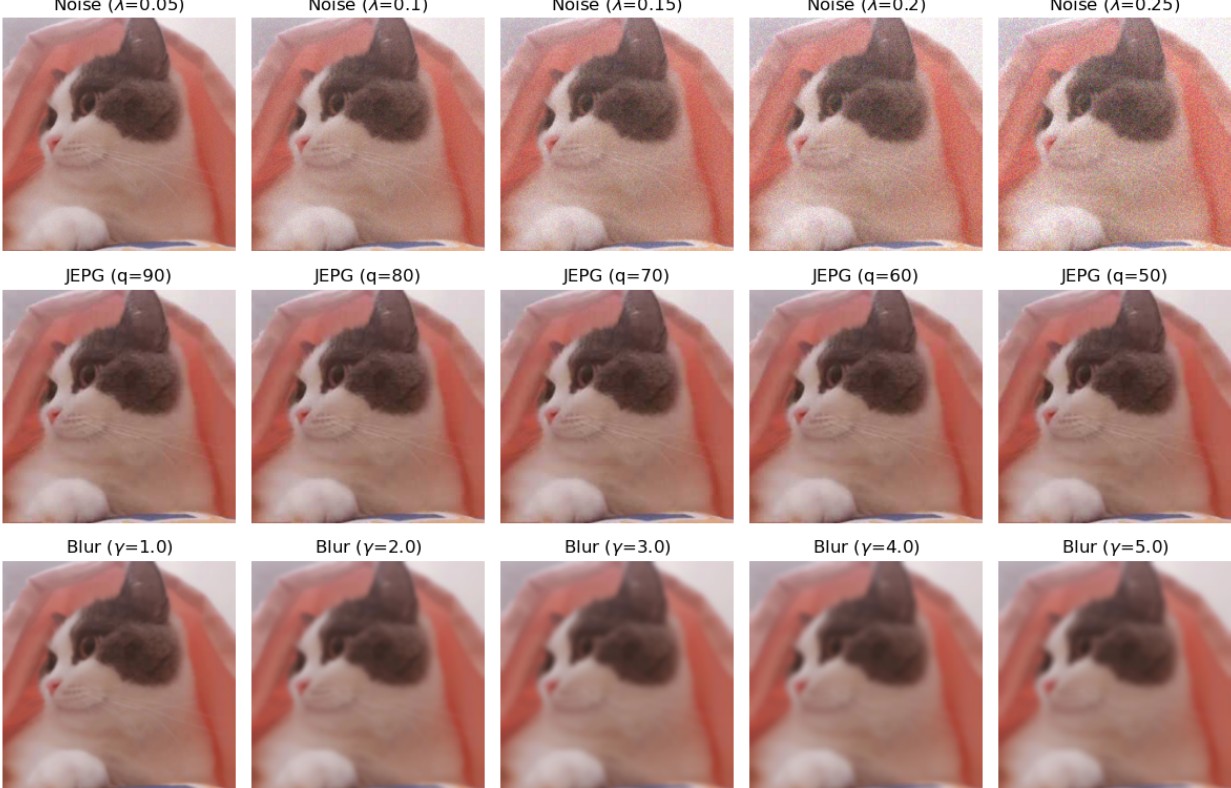

Figure 11: **Display of Perturbed Images.** The first row shows the images perturbed by Gaussian noise with different corruption severities $\eta$. The second row shows the JPEG compressed images with various qualities and the bottom row shows the Gaussian blurred images.

