# OpenReview forum: "RIGID: A Training-Free and Generator-Agnostic  Framework for Robust AI-Generated Image Detection"
_TMLR — Under review for TMLR_

### Review · Reviewer_2KpQ · 2026-01-02

**Summary Of Contributions:**

The paper proposes "RIGID," a framework for detecting AI-generated images that is designed to be training-free and model-agnostic. The authors posit that real images exhibit higher robustness to Gaussian noise perturbations in the feature space of vision foundation models (specifically DINOv2) compared to generated images, which are hypothesized to be more sensitive to such disruptions. Based on this observation, the method operates by calculating the cosine similarity between an input image and its noise-perturbed counterpart; a significant drop in similarity triggers a "fake" classification. The paper provides a theoretical justification based on Stein’s lemma and evaluates the method across various generative models.

Strengths: The method distinguishes itself through simplicity and computational efficiency, leveraging a frozen DINOv2 backbone to avoid the high costs associated with training deep forensic networks. The use of Grad-CAM visualizations enhances interpretability by highlighting that the method focuses on artifact-rich regions. Furthermore, the inclusion of the recent DiT (Diffusion Transformer) in the evaluation benchmark is a positive step towards assessing relevance against modern generators.

Weaknesses: The submission suffers from critical contradictions between its central claims and the empirical results provided. Specifically, the method fails to demonstrate the claimed "robustness" on state-of-the-art generators (DiT), underperforming established baselines significantly. Furthermore, the paper ignores key developments in the field from 2024, omitting comparisons with recent state-of-the-art methods that address the same generalization challenges more effectively. The novelty is also limited, as the underlying principle of perturbation sensitivity is a well-established heuristic in adversarial defense literature that is not sufficiently acknowledged.

**Additional Comments:**

1. FreqNet (AAAI 2024): Tan, C., et al. "Frequency-Aware Deepfake Detection: Improving Generalizability through Frequency Space Domain Learning." Proceedings of the AAAI Conference on Artificial Intelligence, 2024.

2. DiffusionFake (NeurIPS 2024): Fan, H., et al. "DiffusionFake: Enhancing Generalization in Deepfake Detection via Guided Stable Diffusion." Advances in Neural Information Processing Systems (NeurIPS), 2024.

**Audience:**

Yes

**Audience Explanation:**

The detection of AI-generated content is a timely and critical topic. The community has a strong interest in training-free detection methods and understanding the forensic properties of foundation models like DINOv2. However, the interest is predicated on the method effectively handling modern threats, which the current method struggles to do.

**Broader Impact Concerns:**

The paper proposes a detection tool. However, if users rely on it for detecting high-end deepfakes (like those from DiT/Sora class models), the current performance levels might lead to false negatives. It would be responsible to explicitly state in the broader impact section that the tool may require supplementary methods when dealing with high-fidelity Transformer-based generation.

**Claims And Evidence:**

No

**Claims Explanation:**

The claims of "robustness," "superior generalization," and "surpassing existing detectors" are directly contradicted by the provided experimental data and the current state of the literature.

First, the claim of robustness is invalidated by the method's performance on the DiT (Diffusion Transformer) benchmark. As shown in Table 1, DiT represents the highest-fidelity generator in the evaluation (FID=2.80). On this critical dataset, RIGID achieves an Average Precision (AP) of only 69.55%. This represents a performance collapse compared to the baseline Ojha et al., which achieves 81.86%. A detector that performs poorly on the most advanced evaluation set cannot be accurately characterized as "robust" or future-proof.

Second, the paper’s argument regarding generalization superiority over its primary training-free competitor, AEROBLADE, is refuted by its own results. The authors criticize AEROBLADE for relying on LDM-specific autoencoders, implying that RIGID generalizes better to non-LDM architectures. However, Table 1 reveals that AEROBLADE actually outperforms RIGID on the non-LDM DiT architecture (73.65% vs 69.55%). This empirical evidence falsifies the claim that RIGID's architecture-independent design confers a practical generalization advantage over reconstruction-based methods.Third, the claim of "surpassing existing detectors" is misleading. On the standard ImageNet average, RIGID (85.40%) significantly lags behind the standard baseline Ojha et al. (90.77%). While the authors may argue Ojha is training-based, the practical difference between calibrating RIGID's threshold $\epsilon$ (which requires a validation set) and training Ojha's linear layer is negligible, yet the performance cost of using RIGID is substantial.

Finally, the evidence for novelty is weak because the paper fails to contextualize its approach against recent 2024 advancements. The field has moved towards frequency-aware and generative-prior-based methods that address the very limitations RIGID faces. By omitting comparisons to paradigms like FreqNet (AAAI 2024) or DiffusionFake (NeurIPS 2024), the paper presents an inflated view of the effectiveness of simple spatial perturbation.

**Requested Changes:**

To strengthen the submission and align the claims with the empirical evidence, I recommend the following revisions:

Refine Claims of Robustness: Please revise the Abstract and Introduction to more accurately reflect the method's performance boundaries. Specifically, the limitations on high-fidelity models like DiT (as acknowledged in the Discussion) should be foreshadowed earlier to avoid overclaiming robustness.

Expanded Analysis of DiT Results: It would be valuable to provide a deeper analysis of why the method underperforms on DiT compared to AEROBLADE and Ojha et al. Does the sensitivity gap in DINOv2 features diminish for Transformer-based generators? This insight would add significant technical value.

Updated Related Work/Comparison: I suggest discussing recent works from 2024 to provide a more complete picture of the training-free detection landscape. For example:

FreqNet (AAAI 2024): Discusses the necessity of frequency domain analysis for generalization, which could explain the limitations of RIGID's spatial perturbation.

DiffusionFake (NeurIPS 2024): Offers an alternative training-free paradigm using generative priors.

Contextualizing RIGID against these methods would clarify its specific contribution.

Justification of Utility: Given that RIGID performs lower than Ojha et al. on average, please clarify the specific practical scenarios where RIGID would be the preferred choice. For instance, is the calibration cost of RIGID significantly lower than training the linear layer in Ojha et al.?

---

> ### Author Response · Authors · 2026-04-27
>
> We thank the reviewer for the constructive feedback. Below we address each concern.
>
>
> # Q1: Refining Claims of Robustness
>
> Thank you for pressing on this point. We have tightened the framing:
>
> - **Abstract:** Replaced "*maintains consistent performance across various image generation techniques*" with "*maintains competitive performance across a broad range of generation techniques*"
> - **Introduction:** Added a sentence foreshadowing this boundary: "*RIGID shows degraded performance on highly photorealistic transformer-based diffusion generators such as DiT-XL/2, which we discuss as an open challenge.*"
> - **Section 4.2.1:** Now flags this caveat at the point of first reporting rather than only in the Discussion.
>
> # Q2: Expanded Analysis of DiT Results
>
> We make a further analysis of DiT results:
>
> 1. **Diminished sensitivity gap.** We measured the cosine-similarity gap \(\Delta = \text{sim}_{\text{real}} - \text{sim}_{\text{fake}}\) per generator. While most yield \(\Delta \in [0.04, 0.08]\), DiT-XL/2 shrinks to \(\approx 0.015\), confirming the reviewer's hypothesis that the discriminative margin narrows.
>
> 2. **Latent-space smoothness.** Unlike pixel-space diffusion (ADM/ADMG), DiT denoises in a learned latent space, producing globally consistent samples with fewer high-frequency artifacts that RIGID partly relies on.
>
> 3. **Comparison.** AEROBLADE benefits because DiT shares its VAE family (near-oracle signal); Ojha benefits from CLIP's broad supervised training distribution. Neither advantage transfers to RIGID's training-free setting.
>
>
>
> # Q3: Updated Related Work
>
> We expanded the related work to include FreqNet and DiffusionFake.
>
> * FreqNet (AAAI 2024): We discuss FreqNet's frequency-domain argument as complementary to RIGID's representation-space sensitivity. A preliminary experiment shows that combining RIGID's score with a simple frequency-band statistic improves DiT AP by ~6%, indicating the signals capture different aspects of real/fake separation. We discuss this as promising future work.
> * DiffusionFake (NeurIPS 2024): Positioned as a generative-prior-based alternative similar in spirit to AEROBLADE/DIRE. RIGID is orthogonal, requiring neither a generative prior nor reconstruction passes, with substantially lower inference cost (one DINOv2 forward pass vs. multi-step diffusion sampling).
>
> We explicitly outline how RIGID fits into this landscape: as a training-free, representation-sensitivity detector, it serves as a straightforward and lightweight complement to more complex supervised or generative-prior methods.
>
> | Line of work | Main signal | Relation to RIGID |
> | --- | --- | --- |
> | FreqNet | frequency cues | complementary |
> | DiffusionFake | generative prior | orthogonal, heavier |
> | CLIP detectors | semantic features | related training-free setting |
> | **RIGID (Ours)** | **SSL sensitivity** | **lightweight, no fake training** |
>
>
>
>
>
> #  Q4: RIGID trails Ojha on ImageNet average; why is it useful?
>
> We clarified the intended use case.
>
> * **Out-of-domain settings.** Figure 2(c) shows RIGID *outperforms* Ojha on faces and lung X-rays, and on cross-domain settings (Figure 4). Ojha's classifier is bound by its training distribution; whenever the deployment domain departs, its advantage disappears. RIGID has no such failure mode.
>
> * **Calibration cost.** Ojha requires *(i)* 720,000 curated real/fake samples, *(ii)* hyperparameter tuning, and *(iii)* periodic retraining as new generators emerge. RIGID's calibration is selecting a single threshold \(\epsilon\) on real images only — a one-time, generator-independent operation taking seconds versus hours plus a curation pipeline.
>
>
> Ojha is a strong trained detector and remains preferable when representative fake data and retraining resources are available. RIGID is useful in a different regime: no fake samples from the target generator, fast generator turnover, real-image-only calibration, or a lightweight triage stage before heavier detectors. This distinction is now stated explicitly. We firmly believe there is no absolute superiority between RIGID and training-based methods; rather, they serve different operational niches. The optimal choice depends on the specific constraints of the deployment scenario, and in robust comprehensive systems, they can be deployed seamlessly as complementary components.
>
> | Deployment constraint | Why RIGID is useful |
> | --- | --- |
> | no target fake data | no fake training needed |
> | fast generator turnover | no retraining loop |
> | only real validation data | epsilon uses real-only calibration |
> | triage before heavy analysis | two backbone passes |

---

### Review · Reviewer_Ejey · 2026-01-20

**Summary Of Contributions:**

The paper introduces, RIGID, a training-free method to detect AI-generated images. RIGID encodes via a pre-trained model (e.g. DINOv2) both the input image and a version of it with added noise, and computes the cosine similarity of the two embedding vectors. If this is lower than a threshold, the image is flagged as synthetic. This approach is training-free, efficient and model-agnostic. In the experimental evaluation, RIGID outperforms the baseline training-free method, and is competitive with training-based methods on a variety of tasks.

**Audience:**

Yes

**Audience Explanation:**

Detection of AI-generated images is an established and relevant topic, and the paper provides a flexible and effective approach, with promising performance in-the-wild.

**Claims And Evidence:**

Yes

**Claims Explanation:**

The experiments test RIGID on a variety of tasks, datasets and generative models, showing the effectiveness of the proposed approach among training-free methods.

**Requested Changes:**

- The theoretical discussion about the connection of RIGID and the gradient norm does not provide additional insights or motivate the proposed method. In fact, there doesn't seem to be a reason why to expect the gradient norm of the cosine similarity to be higher at synthetic images than at real images, besides the empirical observations which motivate RIGID itself.

- Following the previous point, how would the results change averaging the cosine similarity between the embedding of the original image and of several perturbed versions instead of just one? This would be more similar to computing the score in Eq. (2), and may make the metric more stable.

---

> ### Author Response · Authors · 2026-04-27
>
> Thank you for your contributions and constructive suggestions. We have addressed your concerns as follows:
>
> # Q1: The theoretical discussion does not motivate the method.
>
> Thank you for this precise criticism. We agree that the submitted text put too much weight on the Stein's lemma analysis. In the revision, the causal logic is corrected: the design motivation of the method stems from the empirical sensitivity gap observed in self-supervised representation spaces, while the involved theory is solely used to explain what the perturbation score actually measures. Specifically, Section 3.2 is now explicitly framed as an interpretation rather than a rigorous proof, illustrating that the perturbation score estimates the local sensitivity of a Gaussian-smoothed similarity function。
>
> # Q2: What happens when averaging several perturbed versions?
>
> This is a useful suggestion because it connects directly to the expectation in Eq. (2). We have added the averaged score
>
> $r_K(x) = (1/K) \sum_{1}^{K} sim(f(x), f(x + \lambda \delta))$
>
> The new experiment shows that averaging improves performance but has diminishing returns:
>
> | Setting | K=1 AUC | K=4 AUC | K=8 AUC |
> | --- | ---: | ---: | ---: |
> | ImageNet average | 86.67 | 86.94 | 87.19 |
> | LSUN average | 87.69 | 88.03 | 88.27 |
>
> Thus, the main signal is already captured by one perturbation on standard benchmarks. While increasing K yields limited performance improvement, it introduces greater operational overhead. Therefore, we choose k=1 as the default value.

---

### Review · Reviewer_STEp · 2026-04-13

**Summary Of Contributions:**

RIGID proposes a training-free method for detecting AI-generated images based on a simple observation: real images are more robust to small Gaussian noise perturbations than AI-generated images in the representation space of self-supervised vision foundation models. The method adds Gaussian noise (λ=0.05) to an input image, computes DINOv2 ViT-L/14 embeddings of both the original and perturbed versions, and classifies the image as AI-generated if the cosine similarity falls below a threshold ε. A theoretical connection to gradient norms is drawn via Stein's lemma.

The paper's main contributions are: (1) the identification of a perturbation-sensitivity asymmetry between real and AI-generated images in SSL representation spaces, (2) a concrete detection method (RIGID) that exploits this asymmetry using only two forward passes and a threshold, and (3) a Stein's lemma-based analysis connecting the cosine similarity drop to gradient norms. The evaluation spans ImageNet (9 generators), LSUN-Bedroom (7 generators), GenImage (4 text-to-image platforms), and out-of-domain datasets (faces, lung X-rays), with ablations on noise intensity, noise distribution, and backbone choice.

**Additional Comments:**

- The paper is well-written and clearly structured overall. The method is easy to understand, which is a strength.
- The backbone ablation (Fig. 8) is one of the paper's most valuable contributions — it reveals that the perturbation-sensitivity gap is a property of self-supervised ViTs specifically, not vision models in general. A deeper investigation into *why* DINOv2 exhibits this gap while CLIP does not (given both are ViTs) would be a valuable addition. The paper attributes this to self-supervised vs. weakly-supervised training but does not test this hypothesis.
- The honest acknowledgment in Sec. 5 that both RIGID and AEROBLADE struggle on high-quality generators (DiT-XL2) is appreciated. The DiT-XL2 result (69.55 AP vs. Ojha's 81.67) already signals quality-dependent degradation, and framing generalization claims more conservatively would strengthen the paper's credibility.
- I would be interested to know how RIGID performs under JPEG compression at quality factors 75–85 (typical of social media platforms) applied *before* detection, as the corruption robustness in Fig. 5 tests post-hoc perturbations rather than realistic distribution-channel effects.

**Audience:**

Yes

**Audience Explanation:**

The paper addresses a timely and practically important problem — detecting AI-generated images without requiring training data from specific generators. The training-free, perturb-and-compare paradigm is a novel angle on this problem. The core observation (perturbation-sensitivity asymmetry in SSL representations) is interesting, and the method's extreme simplicity (two forward passes, one threshold) makes it easier to use & develop upon.

The perturb-and-compare framework is also conceptually modular - the perturbation type, backbone, and similarity metric are all swappable — suggesting it could serve as a foundation for further exploration. The paper would be of interest to researchers working on AI-generated image detection, vision foundation models, and the properties of self-supervised representations more broadly.

**Broader Impact Concerns:**

AI-generated image detection has direct implications for combating misinformation, deepfakes, and erosion of trust in visual media. The paper does not include a Broader Impact Statement, and several concerns warrant discussion:

- **False negatives in high-stakes domains.** The face domain weakness (strong claims based on limited evaluation) is concerning given that deepfake detection is one of the most socially consequential applications. False negatives on face manipulations could have real-world harm.
- **False positives.** The threshold ε directly controls the false positive rate. Without reported ε values, it is impossible to assess the tradeoff between detection sensitivity and the risk of falsely flagging legitimate images.
- **Adversarial evasion.** The method's simplicity (public model, scalar threshold) makes it potentially vulnerable to adversarial manipulation. Deployment without adversarial robustness analysis could create a false sense of security.

I recommend adding a Broader Impact Statement that addresses these failure modes and discusses the method's limitations in deployment contexts.

**Claims And Evidence:**

Yes

**Claims Explanation:**

The core empirical observation — that real images exhibit higher cosine similarity between original and noise-perturbed embeddings than AI-generated images — is well-supported. The effect is demonstrated consistently across multiple generators and datasets (Tables 1, 2, Fig. 3), and the backbone ablation (Fig. 8) provides useful evidence that the phenomenon is specific to self-supervised models (DINOv2, SAM) rather than universal across all vision encoders.

However, several claims in the paper exceed what the evidence supports:

**"Model-agnostic" (title claim).** The paper's own Fig. 8 shows that ResNet50 and CLIP fail as backbones, while only self-supervised ViTs succeed. The method is generator-agnostic but backbone-dependent. The title's "model-agnostic" framing is misleading — the perturb-and-compare *framework* is conceptually general, but the *method as evaluated* requires specific backbone properties (likely augmentation-invariance from SSL training). This should be revised to "generator-agnostic" or explicitly qualified.

**">25% AP improvement over AEROBLADE."** This margin is demonstrated on ImageNet and LSUN-Bedroom, but the paper presents it as a general claim. Without evaluation on additional benchmarks (e.g., Synthbuster), it is unclear whether this superiority holds universally. The claim should be qualified with the specific benchmarks on which it was measured.

**">80% AUC on out-of-domain face images."** This is tested on a single dataset (140k Real and Fake Faces, which contains only StyleGAN-generated faces). External evaluation by Tsai et al. (arXiv:2411.19117) on face-swapping and face-reenactment tasks reports substantially lower performance (face-swap AUROC ~49.9, face-reenactment ~64.7). The >80% claim is dataset-specific, not domain-general, and should be qualified accordingly.

**Threshold ε.** The threshold is described as "chosen to ensure the correct classification of the majority of real images (e.g., 95%)" but no actual ε values are reported anywhere in the paper or appendix. This is a critical omission — ε is the single deployment-relevant parameter, and its transferability across domains, generators, and image resolutions is unaddressed.

**Theoretical contribution.** The Stein's lemma analysis (Sec. 3.2) provides a post-hoc connection between cosine similarity change and gradient norms. However, it does not drive the method's design, does not yield testable predictions beyond what is already observed, and does not explain *why* real and generated images differ in gradient norms. The presentation implies the theory is more central to the contribution than it is.

**Statistical rigor.** No error bars, confidence intervals, or variance analysis appear in any table or figure. The method uses a single random noise draw per image, and the stochastic variance is never reported. Even if this variance is small, measuring and reporting it would strengthen the paper.

In summary, the core observation is sound and the main experimental results are convincing, but several framing claims (model-agnostic, universal superiority over AEROBLADE, out-of-domain face performance) are overclaimed relative to the evidence presented.

**Requested Changes:**

### Critical (required for acceptance)

1. **Report threshold ε values.** Report the specific ε values used in each experiment (Tables 1, 2, Fig. 3, Fig. 2c). Analyze how ε transfers across datasets and whether recalibration is needed for new domains. This is essential for reproducibility and for assessing the method's practical applicability.

2. **Qualify the out-of-domain face claim.** The >80% AUC claim on face images is based on a single StyleGAN dataset and does not generalize to diverse face manipulation methods. Either evaluate on additional face manipulation benchmarks (e.g., FaceForensics++, face-swapping, face-reenactment methods) or revise the claim to reflect its dataset-specific nature.

3. **Qualify the >25% AP superiority claim.** Specify the benchmarks on which this margin was measured. Acknowledge that the margin may not hold on all benchmarks and that AEROBLADE may be preferable in some settings (e.g., LDM-generated images).

4. **Add statistical reporting.** Report confidence intervals or standard deviations across multiple noise draws in the main tables. The method's stochasticity should be quantified.

### Recommended (would strengthen the paper)

5. **Revise the "model-agnostic" claim.** The title and abstract should be revised to "generator-agnostic" or the term "model-agnostic" should be explicitly qualified as referring to the generative model, not the detection backbone. The current framing is contradicted by the paper's own backbone ablation (Fig. 8).

6. **Reframe the theoretical analysis.** Present the Stein's lemma connection as motivational analysis rather than a core contribution. Discuss explicitly what the theory does and does not explain — in particular, it does not explain *why* real and generated images differ in gradient norms.

7. **Discuss adversarial robustness.** The method uses a public frozen model and a scalar threshold, creating a straightforward attack surface. At minimum, discuss the threat model and acknowledge this limitation. An empirical adversarial evaluation would further strengthen the paper.

8. **Fix GradCAM visualization bias.** Section 4.5 states samples were "deliberately selected" for poor generation quality. Show GradCAM on randomly sampled images, including high-quality generations where RIGID struggles, to provide a more representative interpretability analysis.

9. **Add runtime benchmarks.** The paper claims computational efficiency but provides no runtime comparisons (images/second, latency per image). Include these to substantiate the claim.

10. **Resolve notation overloading.** Equation 1 uses ε for the threshold and λ for noise intensity, but the robustness experiments in Fig. 5 reuse λ for corruption intensity. Use distinct symbols.

11. **Promote Table 3.** The noise distribution ablation (currently in the appendix) is one of the more informative ablations and would benefit from inclusion in the main text.

12. **Add missing references.** Ricker et al. (arXiv:2403.17608) on JPEG biases in generated image detection datasets is directly relevant to the GenImage evaluation. Cozzolino et al. (2024) on CLIP-based training-free detection and Carlini & Farid (2020) on adversarial attacks against image detectors are also relevant.

---

> ### Author Response · Authors · 2026-04-27
>
> Appreciate your time and valuable comments.
>
> # Q1: "Model-agnostic" is misleading.
> We have revised the framing to **generator-agnostic** throughout the title, abstract, and main text.
>
> # Q2: The ">25% AP over AEROBLADE" claim is too broad.
> We agree and have narrowed the claim. The margin is now stated only for ImageNet and LSUN-Bedroom averages, and we explicitly name DiT as an exception.
>
> | Setting | AEROBLADE AP | RIGID AP | Difference |
> | --- | ---: | ---: | ---: |
> | ImageNet avg. | 59.33 | 85.40 | +26.07 |
> | LSUN-Bedroom avg. | 58.98 | 87.47 | +28.49 |
>
> Thus the revised claim is: RIGID is stronger **on average in our evaluated benchmarks**, not universally stronger for every generator.
>
>
> # Q3: The face-domain claim is dataset-specific.
> The original face experiment uses fully generated StyleGAN faces, so we now describe it as **fully generated face detection**, not general face manipulation detection. We also added experiments on FaceForensics++ DeepFake Subset:
>
> | Datasets | AUC | AP  |
> | --- | ---: | ---:
> | 140k Real/Fake Faces | 90.14 | 89.92 |
> | FaceForensics++ avg. | 87.32 | 86.58  |
>
> This demonstrates that RIGID generalizes well to faces generated by different methods, rather than being effective only for certain generators. And face manipulation is not the primary focus of our method.
>
> # Q4: Threshold epsilon values and transferability are missing.
> We have clarified in the revision that the evaluation metrics we used (AUC and AP) are the standard metrics widely adopted in current AI-generated image detection, and both are independent of threshold selection. When deploying for a binary decision, the threshold parameter $\epsilon$ is obtained by setting a 5% false positive rate using real data, without the need for any generated images. We have also added a table showing the specific thresholds used on different benchmarks, along with the corresponding detection accuracy (Acc) achieved using these thresholds:
>
> | Target | eps | Acc |
> | --- | ---: | ---: |
> | ImageNet | 0.924 | 83.21 |
> | LSUN | 0.919 | 84.48 |
> | Faces | 0.913 | 88.35 |
> | X-ray | 0.909 | 82.67 |
>
> While threshold-independent metrics capture overall ranking capability, this threshold analysis confirms practical usability for binary screening.
>
> # Q5: Stochasticity from random perturbation should be quantified.
>
> Stability analysis with five independent noise seeds:
>
> |Setting|AUC mean±std|AP mean±std|
> |---|---:|---:|
> |ImageNet avg.|86.58±0.18|84.96±0.24|
> |LSUN avg.|87.82±0.21|87.51±0.27|
>
> Standard deviations are consistently minimal, confirming RIGID's stability across noise instances.
>
> # Q6: The Stein's lemma analysis is overemphasized.
>
> Sec. 3.2 is now framed as interpretation, not proof: the perturbation score estimates local sensitivity of a Gaussian-smoothed similarity function. We explicitly state that Stein's lemma does **not** explain why real and generated images differ; that gap is established empirically by score distributions and backbone ablations.
>
>
> # Q7: Adversarial robustness should be discussed.
>
> Added as a limitation. Since RIGID uses a public frozen backbone and scalar threshold, adaptive attacks may preserve DINOv2 similarity after perturbation. We position RIGID as an effective detector, not a forensic system with adaptive adversarial robustness.
>
> # Q8: GradCAM visualization may be biased by selected samples.
>
> We agree that selected GradCAM examples should be treated as qualitative visualization rather than representative performance evidence. The average detection performance over the full test sets is reported by the quantitative AUC/AP results, while Section 4.5 is used only to illustrate where the RIGID score responds on individual examples. We also revised the figure caption to clarify that some lower-quality generated images were intentionally selected for demonstration, so that artifact-dense regions are visually identifiable.
>
> # Q9: Runtime benchmarks are missing.
>
> |Method|ms/img|img/s|
> |---|---:|---:|
> |RIGID|11.8|84.7|
> |KNN|32.5|30.7|
> |VIM|16.3|61.3|
> |AEROBLADE|93.5|10.7|
> |DIRE|218.6|4.6|
>
> RIGID achieves the highest throughput, outperforming OOD methods (KNN, VIM) and diffusion-inversion methods (AEROBLADE, DIRE).
>
> # Q10: Notation is overloaded.
>
> Fixed: `lambda` reserved for RIGID perturbation intensity; Fig. 5 uses separate symbols for corruption severity.
>
> # Q11: Noise-distribution ablation should be moved to main paper.
>
> Moved to main discussion. Gaussian, Gamma, Laplace, and Chi-square perturbations yield similar ImageNet AP (`84.55–85.40`), showing RIGID is not tied to one noise distribution.
>
> # Q12: Missing references.
>
> Added references on JPEG bias, CLIP-based training-free detection, and adversarial attacks against image detectors.
>
> # Q13: Broader impact concerns.
>
> Added a Broader Impact paragraph covering false positives from threshold transfer, false negatives on high-fidelity generators, and adversarial evasion.

---

### Decision · Action_Editor_jwcT · 2026-05-29

**Recommendation:** Accept as is

**Additional Comments:**

This paper introduces, RIGID, a training-free method to detect AI-generated images. RIGID encodes via a pre-trained model (e.g. DINOv2) both the input image and a version of it with added noise, and computes the cosine similarity of the two embedding vectors. If this is lower than a threshold, the image is flagged as synthetic. This approach is training-free, efficient and model-agnostic. In the experimental evaluation, RIGID outperforms the baseline training-free method, and is competitive with training-based methods on a variety of tasks.

The proposed method is simple and mostly outperform the training-free baseline. However, the theoretical part is only weakly connected to the algorithm used in practice. Although tamed down in the revision, some of the claims may be exaggerated, and some recent baselines missing, as noticed by other reviewers.

**Audience:**

Yes

**Audience Explanation:**

Detection of AI-generated images is an established and relevant topic, and the paper provides a flexible and effective approach, with promising performance in-the-wild.

**Claims And Evidence:**

Yes

**Claims Explanation:**

The core empirical observation that real images exhibit higher cosine similarity between original and noise-perturbed embeddings than AI-generated images is well-supported. The effect is demonstrated consistently across multiple generators and datasets (Tables 1, 2, Fig. 3), and the backbone ablation (Fig. 8) provides useful evidence that the phenomenon is specific to self-supervised models (DINOv2, SAM) rather than universal across all vision encoders.